# Screening for Occult Transthyretin Amyloidosis in Patients with Severe Aortic Stenosis and Amyloid Red Flags

**DOI:** 10.3390/jcm13030671

**Published:** 2024-01-24

**Authors:** Aiste Monika Jakstaite, Julia Kirsten Vogel, Peter Luedike, Rolf Alexander Jánosi, Alexander Carpinteiro, Christoph Rischpler, Ken Herrmann, Tienush Rassaf, Maria Papathanasiou

**Affiliations:** 1Department of Cardiology and Vascular Medicine, West German Heart and Vascular Center, University Hospital Essen, Hufelandstrasse 55, 45147 Essen, Germany; 2Department of Hematology and Stem Cell Transplantation, West German Tumor Center, University Hospital Essen, Hufelandstrasse 55, 45147 Essen, Germany; 3Institute of Molecular Biology, University of Duisburg-Essen, Hufelandstrasse 55, 45147 Essen, Germany; 4Department of Nuclear Medicine, Klinikum Stuttgart, Kriegsbergstrasse 60, 70174 Stuttgart, Germany; 5Department of Nuclear Medicine, University Hospital Essen, Hufelandstrasse 55, 45147 Essen, Germany

**Keywords:** amyloidosis, ATTR amyloidosis, cardiomyopathy, heart failure, valvular heart disease

## Abstract

Aims: The optimal strategy to identify transthyretin-type cardiac amyloidosis (ATTR-CA) in patients with aortic stenosis (AS) is still unclear. This study aimed to investigate if targeted screening for ATTR-CA in patients with severe AS and amyloid red flags is associated with higher detection rates. Methods: The study prospectively enrolled patients ≥65 years with severe AS. Patients who fulfilled ≥1 major (carpal tunnel syndrome (CTS), ruptured biceps tendon, spinal stenosis, N-terminal pro B-type natriuretic peptide ≥1000 pg/mL, cardiac troponin >99th percentile) or ≥2 minor criteria (diastolic dysfunction ≥2 grade/lateral e’ <10 cm/s, atrial fibrillation, atrioventricular conduction disease/pacemaker) received bone scintigraphy and biochemical analysis for light chain amyloidosis. Hypertensive patients (>140/90 mmHg) and those with interventricular septal thickness (IVSd) ≤13 mm were excluded. Results: Overall, 264 patients were screened, of whom 85 were included in the analysis. Tracer uptake Perugini grade ≥1 was detected in nine patients (11%). An endomyocardial biopsy was additionally performed in four of nine patients, yielding a prevalence of 7% (n = 6). All patients with dual AS-ATTR were male. Syncope was more commonly reported in AS-ATTR patients (50% vs. 6%, *p* = 0.010), who also tended to have more severe hypertrophy (IVSd of 18 vs. 16 mm, *p* = 0.075). Pericardial effusion and CTS were more common in patients with dual pathology (67% vs. 8%, *p* < 0.001, and 83% vs. 24%, *p* = 0.003, respectively). Conclusion: Targeted screening for ATTR-CA in patients with AS and amyloid red flags does not yield higher detection rates than those reported previously in all comers with AS.

## 1. Introduction

Degenerative aortic stenosis (AS) is a common disease in the elderly, affecting over 4% of individuals ≥70 years of age [1]. Transcatheter aortic valve replacement (TAVR) has transformed the treatment and outcomes of AS in high-risk and inoperable patients with an increasing interest to expand its application to all-risk patients [2]. According to previous studies, transthyretin-type cardiac amyloidosis (ATTR-CA) has a prevalence of 8% to 16% among all comers with severe AS undergoing TAVR and up to 4% in patients undergoing surgical aortic valve replacement (SAVR) [3,4,5,6,7,8]. Autopsy data report ATTR-CA in 25% of patients aged ≥85 years, suggesting that the prevalence may increase with age [9]. 

The diagnosis of CA in patients with severe AS remains challenging mainly due to the lack of disease-specific biomarkers and echocardiographic similarities between severe AS and CA. Routine screening for CA in all patients with AS using bone scintigraphy is not feasible and cost-effective in everyday clinical practice. On the other hand, screening only the subgroups of the AS population with a higher probability of ATTR-CA (e.g., low flow, low gradient AS) may lead to underdiagnosis. Recently, a scoring system was proposed for the identification of amyloidosis in AS patients. This included clinical (history of carpal tunnel syndrome (CTS), 3 points; age ≥ 85 years, 1 point), echocardiographic (left ventricular septal wall thickness ≥18 mm, 1 point; E/A ratio > 1.4, 1 point), electrocardiographic (right bundle branch block, 2 points; Sokolow–Lyon index < 1.9 mV, 1 point), and biomarker (high-sensitivity troponin T > 20 ng/L, 1 point) indices. Sensitivity for scores ≥2 and ≥3 was 94% and 72%, and specificity 52% and 84%, respectively; thus, the optimal screening strategy to precisely identify patients with AS-ATTR remains unclear [7]. Screening for amyloid red flags in at-risk cohorts is recommended by international guidelines [10,11]. The predictive value of numerous amyloid red flags has been demonstrated in previous studies across various patient cohorts. Conditions such as CTS and biceps tendon rupture are common red flags that have been shown to occur 5–15 years before the manifestation of CA [12,13]. Similarly, peripheral neuropathy is a common neurological disorder with different etiologies and can also present as the initial symptom of amyloidosis [14,15]. The current study aimed to test the hypothesis that a screening algorithm based on known amyloid red flags leads to higher detection rates of ATTR-CA in patients with severe AS.

## 2. Methods

### 2.1. Study Design

The study prospectively enrolled patients with severe degenerative AS referred to our center for TAVR. Patients were eligible for the study if aged ≥65 years old and fulfilled at least one major or two minor prespecified risk criteria for ATTR-CA. Major criteria included: (1) CTS, (2) a ruptured biceps tendon, (3) spinal stenosis, (4) N-terminal pro B-type natriuretic peptide (NT-proBNP) ≥1000 pg/mL, and (5) high-sensitivity cardiac troponin I (hs-cTnI) >99th percentile upper reference limit. As minor criteria, we defined the following: (1) second or higher grade diastolic dysfunction, or e’ velocity in pulsed wave Doppler of the lateral mitral valve annulus <10 cm/s, (2) atrial fibrillation, and (3) atrioventricular conduction disease or history of conduction disorders warranting pacemaker implantation. Exclusion criteria encompassed arterial hypertension at admission with resting arterial blood pressure ≥140/90 mmHg, interventricular septal thickness ≤13 mm, inability to provide informed consent, or inability to perform scintigraphic imaging for other reasons (e.g., frailty, dementia). All study participants provided written informed consent. The investigation conforms with the principles outlined in the *Declaration of Helsinki* and received approval from the local ethics committee. The study is registered at ClinicalTrials.gov (registry number NCT05693376).

### 2.2. Study Procedures

Patients who fulfilled the inclusion criteria received technetium-99m-labeled 3,3-diphosphono-1,2-propanodicarboxylic acid (^99m^Tc-DPD) scintigraphy with single-photon emission computed tomography/low-dose computed tomography (SPECT/CT) of the chest and biochemical analysis of serum and urine for monoclonal gammopathy/light chain amyloidosis (Figure 1). The latter included serum and urine immunofixation, serum electrophoresis, and serum free light chain assay. Both scintigraphy and biochemical analysis were performed during the index hospitalization for TAVR or shortly thereafter but not later than 3 months post-TAVR. ^99m^Tc-DPD bone scintigraphy was performed using a hybrid SPECT/CT gamma camera (Symbia T2 and Symbia Intevo, Siemens Medical Solutions AG, Erlangen, Germany) with low-energy high-resolution collimators following the weight-adjusted administration of ^99m^Tc-DPD (Dupharma A/S, Kastrup, Denmark). The protocol consisted of early (1 h) and late (3 h) planar whole-body imaging with thoracic SPECT/CT image acquisition after 3 h. The visual Perugini grading system was used for the evaluation of myocardial tracer uptake on planar images with the following classification: 0 = no myocardial uptake, 1 = myocardial uptake less than the rib uptake, 2 = myocardial uptake equal to rib uptake, 3 = myocardial uptake greater than the rib uptake [16]. The heart-to-contralateral lung (H/CL) ratio (ratio of heart regions of interest to contralateral lung region of interest mean counts) was used for quantitative analysis [17]. The diagnosis of ATTR-CA was established according to current guidelines in the case of Perugini grade 2 or 3 uptake with normal serum free light chains kappa and lambda and free light chain ratio and absence of monoclonal band on serum and urine immunofixation [10].

Endomyocardial biopsy was additionally performed in the case of Perugini grade 1 uptake or in patients with abnormal immunofixation or free light chain assays. Endomyocardial biopsies were performed in the supine position via the common femoral vein using the standard Seldinger canulation technique and a long non-flexible sheath. A minimum of 4 samples were obtained from the right ventricular (RV) septum under fluoroscopic guidance using a flexible bioptome. Genetic testing was performed using genomic DNA from peripheral blood in patients with ATTR-CA. The exons of the transthyretin (TTR) gene were amplified and sequenced by single-gene Sanger sequencing. Further laboratory analyzes were performed as per institutional protocol prior to TAVR and included NT-proBNP, hs-cTnI, complete blood count, serum albumin, and renal and liver function testing. Electrocardiograms were recorded before and within 24 h after TAVR. The Sokolow–Lyon index was calculated as the sum of the S-wave in lead V1 and the tallest R-wave in lead V5 or V6 [18]. Low voltage was defined as a QRS amplitude of ≤0.5 mV in all peripheral leads [19]. The voltage/mass ratio (VMR) was calculated by dividing the value of the Sokolow–Lyon index on ECG by the LV mass index (LVMI) on echocardiography. Transthoracic echocardiography for evaluation of AS severity, concomitant valvular pathologies, measurements of cardiac diameters, and systolic and diastolic function were performed in accordance with the latest recommendations [20,21,22]. Qlab 10 software (Philips Electronics, Eindhoven, the Netherlands) was used for offline strain analysis. Apical two-, three-, and four-chamber views were obtained and stored for the analysis of LV global longitudinal strain (GLS).

### 2.3. Statistical Analysis

Continuous variables were summarized as the mean ± standard deviation or median (interquartile range (IQR)) and categorical variables as counts and percentages. Continuous data were evaluated for normality of distribution with the Shapiro–Wilk test and by inspection of the plots. The two-sided *t*-test was used for comparison of continuous, normally distributed data, otherwise the non-parametric Mann–Whitney U test. The x^2^ test was used to test the association between categorical variables. The Kaplan–Meier method was used for survival analysis. The level of significance was set at 0.05. All analyzes were performed using SPSS Version 28 (IBM Corp., Armonk, NY, USA).

## 3. Results

### 3.1. Study Population

During a 24-month period, 101 of 264 consecutive patients met the study criteria and provided written consent. We excluded 15 patients who fulfilled the study inclusion criteria but did not undergo bone scintigraphy for logistical reasons (COVID-19 related restrictions, tracer availability issues) and one patient who died before the scheduled scintigraphy. The analytical cohort included 85 patients (Figure 2). The majority of patients fulfilled at least one major criterion for study inclusion, and 42% had both two minor and at least one major criteria. Table 1 demonstrates the distribution of the inclusion criteria. The most commonly documented criterion was NT-proBNP >1000 pg/mL (84%), followed by CTS (28%). Only 3% of patients were included in the study based only on the combination of two minor criteria.

### 3.2. Baseline Characteristics

Baseline clinical characteristics are demonstrated in Table 2 and Table 3. The mean age was 82 years, and 60% of the patients were male. Study patients were highly symptomatic (88% with NYHA class ≥3) and carried a high burden of comorbidities. The median EuroSCORE II was 5.27 (IQR: 3.88–8.44). After evaluation by the heart team, 79 (93%) patients underwent TAVR. Six patients (7%) who were initially referred for TAVR underwent surgical aortic valve replacement (SAVR). The most common AS phenotype was classical high-gradient AS (64%), followed by low-flow, low-gradient AS with reduced EF (20%), low-flow, low-gradient AS with preserved EF (15%), and normal-flow, low-gradient AS with preserved EF (1%).

### 3.3. The Prevalence of CA

^99m^Tc-DPD scintigraphy detected myocardial uptake (Perugini grade ≥1) in nine patients (11%). Grade 1 was detected in two, grade 2 in two, and grade 3 in five patients. An endomyocardial biopsy was performed in four of nine patients to confirm or rule out amyloidosis according to the current guidelines. In three of these patients no amyloid was found by biopsy, yielding an ATTR-CA prevalence of 7%. Patients with ATTR-CA had a significantly higher H/CL ratio (1.2 vs. 2.7, *p* < 0.001). There were no differences in the amount of radioactivity used (Table 4). Monoclonal gammopathy or a pathologic free light chain assay were detected in 17 (20%) patients who underwent further evaluation by hematologists with expertise in amyloidosis and plasma cell diseases. Light chain amyloidosis was excluded clinically or via biopsy/bone marrow studies in all cases. All but one patient consented to TTR gene sequencing. In all five cases, wild-type ATTR-CA was diagnosed.

### 3.4. Clinical Features of Lone AS vs. AS-ATTR

Patients with dual AS-ATTR pathology were male (100% in AS-ATTR vs. 57% in lone AS, *p* = 0.011). No differences were observed between ATTR amyloidosis and non-amyloidosis patients regarding the NYHA functional class (Table 2). Syncope was reported more frequently in AS-ATTR patients (50% vs. 6%, *p* = 0.010). No differences were observed between ATTR and AS-ATTR patients regarding the above-mentioned major and minor criteria/amyloid red flags, except for CTS, which was more common among patients with amyloidosis (83% vs. 24%, *p* = 0.003). Levels of BNP were significantly higher in AS-ATTR patients (251.6 pg/mL (IQR: 153–386) vs. 589.9 pg/mL (IQR: 294–1007), *p* = 0.035) and a trend toward higher levels of NT-proBNP (2449 pg/mL (IQR: 1256–5274) vs. 4081.5 pg/mL (IQR: 1868.8–18771.5), *p* = 0.181) and troponin I (18 pg/mL (IQR: 10.8–41.3) vs. 47 pg/mL (IQR: 10.8–62.5)) was shown in patients with ATTR-CA compared to those without amyloidosis (Table 3).

Regarding the electrocardiographic features, no differences were observed between the two groups. None of the AS-ATTR patients met electrocardiographic criteria for low voltage. VMR and Sokolow–Lyon, among other indices, did not differ between the groups. There were no differences in hemodynamic measures of AS (aortic valve area 0.7 cm^2^ vs. 0.63 cm^2^, *p* = 0.301, mean pressure gradient 40.54 mmHg vs. 31.08 mmHg, *p* = 0.184, peak aortic valve velocity 4.08 m/s vs. 3.44 m/s, *p* = 0.062). The prevalence of either classical low-flow, low-gradient AS with reduced EF (19% vs. 33.3%, *p* = 0.425) or low-flow, low-gradient AS with preserved EF (15.2% vs. 16.7%, *p* = 0.924) was similar among the two groups. Pericardial effusion was more common in the AS-ATTR group (8% vs. 67%, *p* < 0.001). There were no differences in the measures of LV systolic function (LVEF 52% vs. 52%, *p* = 0.992, MCF 27.38% vs. 24.12%, *p* = 0.524, GLS −12.98% vs. −12.05%, *p* = 0.624) and right ventricular (RV) function (TAPSE 20.24 mm vs. 19.5 mm, *p* = 0.749, RV s’ 10.43 cm/s vs. 9.4 cm/s, *p* = 0.403). Patients with dual AS-ATTR had higher septal thickness, but this did not reach clinical significance (IVST 1.56 cm vs. 1.75 cm, *p* = 0.075, PWT 1.28 cm vs. 1.37 cm, *p* = 0.503, LVMI 149.72 g/m^2^ vs. 162.08 g/m^2^, *p* = 0.473, RWT 0.55 vs. 0.6, *p* = 0.562). The measures of diastolic function did not differ significantly (Table 3).

### 3.5. Periprocedural Complications and Mortality

Complications between lone AS and dual AS-ATTR occurred with similar rates, including new-onset conduction disorders requiring pacemaker implantation that were observed in eight subjects without amyloidosis (11%). One-third of AS-ATTR patients had a pacemaker implanted before the valve procedure (Table 5). Thirty-day survival after TAVR was 100%. Three patients died within the first 30 days post-SAVR. During a median follow-up of 18 (10–23) months, nine (11%) patients (all with lone AS) died. The all-cause mortality of lone AS vs. AS-ATTR was 12% and 0%, respectively, *p* = 0.228 (Figure 3).

## 4. Discussion

### 4.1. The Prevalence of ATTR-CA in Patients with Severe AS

The diagnosis of concomitant amyloidosis in patients with severe AS is of paramount importance and enables the implementation of disease-modifying therapies in addition to valve replacement to achieve optimal long-term outcomes. Previous studies reported the coexistence of amyloidosis in up to 16% of patients with severe AS. Since then, multiple attempts have been made to characterize this population and derive scoring systems for the identification of dual pathology [3,4,5,7,23,24]. A previously proposed score for discrimination of lone AS from AS with concomitant CA was shown to have high specificity but low sensitivity, thus limiting its broader clinical implementation [7]. The current study shows that despite prospective systematic screening for amyloid red flags, an approach that is recommended by international guidelines and consensus documents for patients older than 65 years old [10,11], the ATTR-CA detection rate in severe AS remains lower than that previously reported in all comers with severe AS. With this approach, fourteen patients with severe degenerative AS should undergo diagnostic work-up for one patient to be diagnosed with ATTR-CA. The low observed ATTR-CA prevalence in our study may be attributed to the slightly younger patient cohort of the current study. The mean age was 82 years, lower than that reported in previous studies (83.6–85 years) [4,6,7]. This hypothesis is also supported by CA screening studies in SAVR patients [3,24] that demonstrated a much lower prevalence, e.g., 6% in a patient cohort with a median age of 75 years. Post-mortem studies confirm a higher prevalence of amyloidosis in octogenarians and further support the age-dependent prevalence of ATTR-CA [9]. Further, it is possible that previous studies on all comers with severe AS overestimated the prevalence of ATTR-CA due to selection bias or false positive scintigraphic findings. The latter was also observed in our study in a patient with grade 2 uptake who was later proven negative for amyloid by an endomyocardial biopsy. It is not known if AS and the accompanying myocardial fibrosis could lead to false-positive DPD scintigraphy, but false-positive studies were previously reported in patients after acute myocardial infarction [25].

### 4.2. The Predictive Value of Amyloid Red Flags

The current findings provide insight into the potential predictive value of clinical manifestations known to serve as amyloid red flags in patients who are thought to carry a higher risk, such as those with severe AS. In this cohort, CTS and NT-proBNP >1000 pg/mL were the two most prevalent criteria associated with ATTR-CA in this cohort; however, the low number of diagnoses does not allow for the derivation of a predictive model. CTS is a common condition with a prevalence of 5% in the general population aged 65–74 years [26]. It has been shown that CTS is associated with a future diagnosis of CA and is diagnosed 10–15 years before cardiac impairment [27]. In an extensive analysis of the Danish national registries, CTS was associated with a 12-fold higher future risk of amyloidosis [28]. Our results corroborate these findings. However, 80% of patients with AS who had CTS had a negative DPD scan, suggesting that other etiologies of CTS may limit its predictive value in older patients. Similar to CTS, a biceps tendon rupture may represent ATTR deposition. It was observed in 33–44% of patients with diagnosed ATTR-CA and occurred 5 years before the diagnosis of cardiac involvement [13,29]. The value of age-related conditions such as spinal stenosis and biceps tendon rupture in discriminating ATTR-CA from lone AS is still unclear, and prospective data regarding their exact prevalence in systemic amyloidosis are lacking. A possible hypothesis is that an isolated soft-tissue type of ATTR-CA may be more prevalent than ATTR with organ involvement, thus limiting the overall predictability of soft-tissue manifestations.

### 4.3. Prognostic Implications of Dual Pathology: AS and ATTR-CA

Long-term prognosis of patients with severe AS and ATTR-CA are not available so far. The coexistence of ATTR-CA has been associated with notably higher rates of HF hospitalization compared to that with lone AS [7,30]. Regarding mortality, inconsistent data have been reported. Several studies have reported similar survival of patients with dual pathology and lone AS [6,7,31]. In a retrospective study on patients with CA with moderate and severe AS, ATTR-CA was associated with higher mortality at one year (56% vs. 20%, *p* < 0.0001). After adjustment for potential confounders, CA remained an independent predictor of all-cause mortality [30]. Nitsche et al. reported a trend toward higher 1-year mortality (25.0% vs. 13.9%, log-rank *p* = 0.05) [7]. TAVR significantly improves outcomes in both lone AS and AS-ATTR patients [6,7,32] but AS-ATTR patients remain more symptomatic at 12 months after TAVR [33], further justifying the need for disease-modifying pharmacotherapies in AS-ATTR. From a pathophysiological point of view, it is possible that TAVR could lead to partial stabilization of amyloidogenesis after the reduction of AS-induced myocardial strain that could favor amyloid tissue infiltration. On the other hand, reducing amyloid load could potentially play a role in the management of AS. Amyloid may contribute to AS progression, as demonstrated in a study where surgically resected heart valve specimens were analyzed, revealing amyloid deposits in 55% of the specimens, with the highest prevalence observed in AS [34]. In our study, we found no difference in all-cause mortality at the 18-month follow up, although the small study sample limits the interpretation of this finding. The ATTR stabilizer tafamidis was prescribed in the ATTR-CA group and may have catalyzed this outcome.

All patients in our study with a dual pathology of AS-ATTR were highly symptomatic (NYHA stage III in all). At the current stage, it is challenging to determine whether screening for ATTR-CA in such highly symptomatic patients is prognostically useful, as therapeutic options are limited. On the other hand, therapies currently in development to induce amyloid removal in ATTR-CA could potentially be used to treat patients in more advanced disease stages. 

Recently, machine learning approaches based on clinical, laboratory, and imaging data have increasingly been employed to suspect CA. Machine learning has enabled the early identification of ATTRv and helped identify patients with neuropathy who should undergo genetic testing for ATTRv [35,36]. Furthermore, an ECG-based tool has effectively suspected CA with an area under the curve of 0.97 [37]. Similarly, a deep learning approach discriminated between amyloidosis and hypertrophic cardiomyopathy [38]. Given the increasing number of patients, automated machine learning approaches could potentially be employed for the early identification of patients with cardiac amyloidosis in the future.

It remains challenging to identify patients with AS who have the highest probability of dual pathology of AS and ATTR-CA. We have demonstrated that the classical amyloidosis features were not of high predictive value in a population of patients with AS. Our findings highlight the unmet need for novel and specific amyloidosis markers in patients with AS. The true prevalence of ATTR-CA in AS should be prospectively investigated in larger studies based on a method that ensures the highest diagnostic accuracy, such as endomyocardial biopsy.

### 4.4. Limitations

This study was conducted at a referral tertiary center, thus inducing a potential selection bias and leading to the underrepresentation of the general population of AS patients. The study had a relatively high dropout rate. The small patient numbers in the AS-ATTR group limit statistical power to allow for a multivariate predictive model. A control group of patients without red flags and a larger study sample would be necessary for this study to be conclusive regarding the true prevalence of ATTR-CA and the value of this red flag-based screening approach. Given the false-negative scintigraphy findings in patients with certain mutations, this could potentially have led to the underestimation of ATTRv, as genetic testing was only performed in patients with positive (Perugini > 1) bone scans, despite its low prevalence in the elderly. We excluded patients with high blood pressure values in whom hypertension-induced hypertrophy may exist. Although patients with ATTR-CA, especially in more advanced disease stages, typically experience low blood pressure, some might still be hypertensive. This could potentially have led to an underdiagnosis of ATTR-CA in this patient cohort. We used a septal thickness cutoff of >13 mm based on the recommendations available at the time when study inclusion was ongoing. However, increasing evidence suggests that in some cases, ventricular hypertrophy might only be mild. Our results should be considered hypothesis generating and emphasize that more precise screening algorithms are needed beyond clinical red flags to effectively identify ATTR-CA in patients with AS.

## 5. Conclusions

Targeted screening for ATTR-CA in patients with AS and amyloid red flags does not yield higher detection rates than those reported previously in unselected AS populations. Unmasking CA in patients with AS using red flags proves to be challenging. A comprehensive approach is needed to enhance diagnostic accuracy. Beyond traditional manifestations linked to amyloidosis, precise profiling of this dual entity is needed to identify specific disease markers. The true prevalence of ATTR-CA in AS remains unclear and has to be investigated in prospective studies and compared with age-matched controls.

## Figures and Tables

**Figure 1 jcm-13-00671-f001:**
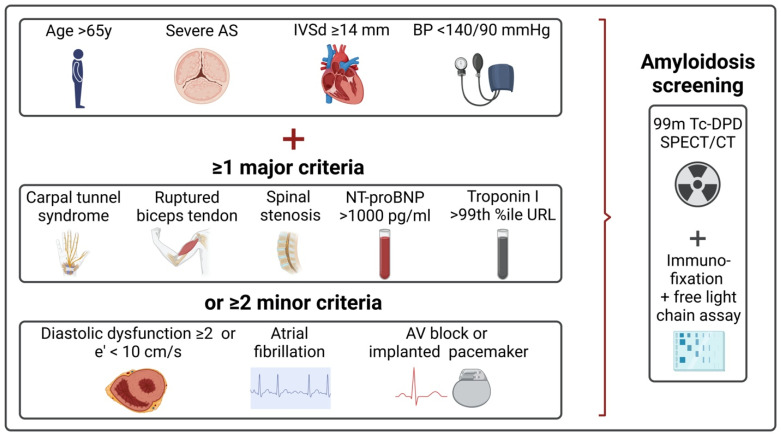
Screening algorithm and study inclusion.

**Figure 2 jcm-13-00671-f002:**
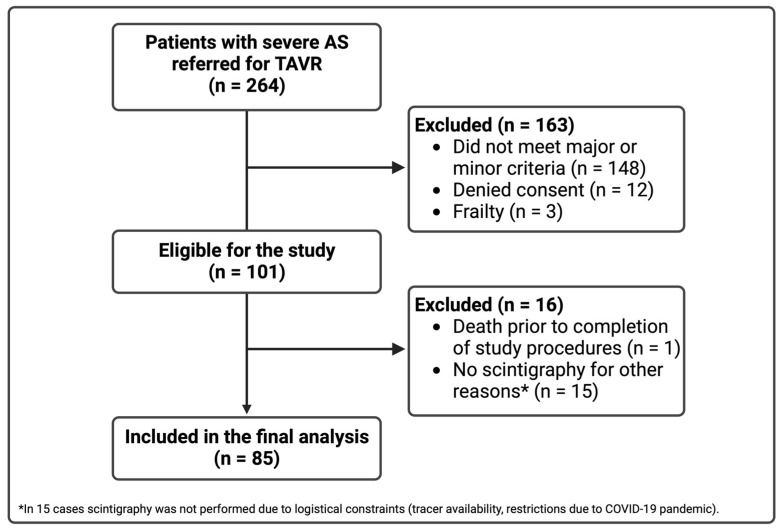
Derivation of the analytic cohort of the study.

**Figure 3 jcm-13-00671-f003:**
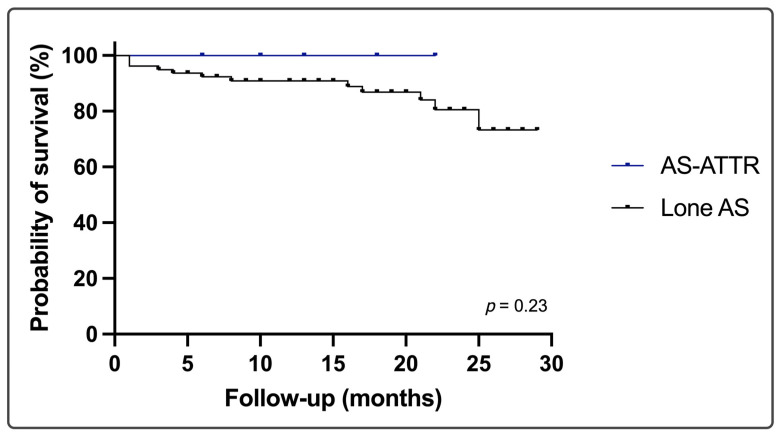
Kaplan–Meier survival analysis for lone AS vs. dual AS-ATTR pathology. AS = aortic stenosis, AS-ATTR = dual pathology of aortic stenosis and transthyretin cardiac amyloidosis.

**Table 1 jcm-13-00671-t001:** Distribution of major and minor criteria for inclusion in the study.

	All n = 85	Lone AS n = 79	AS and ATTR-CA n = 6	*p*-Value
**Major criteria**				
CTS, n (%)	24 (28)	19 (24)	5 (83)	0.003
Biceps tendon rupture, n (%)	16 (19)	15 (19)	1 (17)	0.887
Spinal stenosis, n (%)	16 (19)	14 (18)	2 (33)	0.379
NT-proBNP > 1000 pg/mL	71 (84)	65 (82)	6 (100)	0.134
Hs-cTnI > 99th percentile	20 (24)	17 (22)	3 (50)	0.142
**Minor Criteria**				
AF, n (%)	50 (58.8)	45 (57)	5 (83)	0.206
Diastolic dysfunction ≥2 grade or lateral e’ <10 cm/s	40 (52)	35 (49)	5 (83)	0.094
Atrioventricular block or previously implanted pacemaker	25 (29)	22 (28)	3 (50)	0.272
**Inclusion reason (criteria)**				
1 major	21 (24.7)	21 (26.6)	0 (0)	0.059
2–3 major	24 (28.2)	23 (29.1)	1 (16.7)	0.493
4–5 major	1 (1.2)	1 (1.3)	0 (0)	0.701
2 minor only	3 (3.5)	3 (3.8)	0 (0)	0.504
2 minor + ≥1 major	36 (42.4)	31 (39.2)	5 (83.3)	0.032

AF = atrial fibrillation, AS = aortic stenosis, ATTR-CA = transthyretin cardiac amyloidosis, CTS = carpal tunnel syndrome, hs-cTnI = high-sensitivity cardiac troponin I, NT-proBNP = N-terminal pro B-type natriuretic peptide.

**Table 2 jcm-13-00671-t002:** Baseline characteristics.

	All n = 85	Lone AS n = 79	AS and ATTR-CA n = 6	*p*-Value
**Age, years**	81.7 ± 5.2	81.5 ± 5.2	84.8 ± 4.2	0.123
**Male sex, n (%)**	51 (60)	45 (57)	6 (100)	0.011
**BMI, kg/m^2^**	27.1 ± 4.9	27.3 ± 0.6	24.2 ± 1.5	0.138
**Euroscore II**	5.3 (3.9–8.4)	5.3 (3.8–8.4)	5.1 (4.4–15.6)	0.498
**NYHA class, n (%)**				0.593
I	1 (1)	1 (1)	0 (0)	
II	9 (11)	9 (11)	0 (0)	
III	73 (86)	67 (85)	6 (100)	
IV	2 (2)	2 (3)	0 (0)	
**History of syncope, n (%)**	8 (9)	5 (6)	3 (50)	0.010
**AS phenotype**				
High gradient, n (%)	54 (64)	51 (65)	3 (50)	0.483
LFLG with reduced EF, n (%)	17 (20)	15 (19)	2 (33)	0.425
LFLG with preserved EF, n (%)	13 (15)	12 (15)	1 (17)	0.924
Normal-flow low-gradient with preserved EF, n (%)	1 (1)	1 (1)	0 (0)	0.701
**Comorbidities**				
Peripheral neuropathy, n (%)	9 (11)	8 (10)	1 (17)	0.638
Pre-interventional pacemaker, n (%)	11 (13)	9 (11)	2 (33)	0.177
Hypertension, n (%)	76 (89)	70 (89)	6 (100)	0.237
Diabetes mellitus, n (%)	19 (22)	18 (23)	1 (17)	0.720
Dyslipidaemia, n (%)	51 (60)	46 (58)	5 (83)	0.201
Atrial fibrillation, n (%)	50 (59)	45 (57)	5 (83)	0.206
Coronary artery disease, n (%)	62 (73)	57 (72)	5 (83)	0.534
Previous stroke, n (%)	6 (7)	6 (8)	0 (0)	0.340
Chronic kidney disease, n (%)	36 (42)	33 (42)	3 (50)	0.696
Anemia, n (%)	30 (35)	27 (34)	3 (50)	0.444

AS = aortic stenosis, ATTR-CA = transthyretin cardiac amyloidosis, BMI = body mass index, EF = ejection fraction, LFLG = low-flow low-gradient, NYHA = New York Heart Association.

**Table 3 jcm-13-00671-t003:** Laboratory, echocardiographic, and electrocardiographic findings prior to TAVR.

	All n = 85	Lone AS n = 79	AS and ATTR-CA n = 6	*p*-Value
**Laboratory testing**				
NT-proBNP (pg/mL)	2518 (1340.5–5319.5)	2449 (1256–5274)	4081.5 (1868.8–18771.5)	0.181
BNP (pg/mL)	265.7 (163.3–420)	251.6 (153–386)	589.9 (294.1–1007.4)	0.035
Hs-cTnI (ng/L)	18 (11–42.8)	18 (10.8–41.3)	47 (10.8–62.5)	0.305
eGFR (mL/min/1.73 m^2^)	56.5 ± 20.8	56.7 ± 20.8	53.7 ± 21.6	0.739
Creatinine (mg/dL)	1.3 ± 0.7	1.3 ± 0.7	1.4 ± 0.5	0.807
Haemoglobin (g/dL)	11.9 ± 1.9	11.9 ± 1.9	11.3 ± 2.1	0.463
Albumin (mg/L)	3.6 ± 0.4	3.6 ± 0.5	3.8 ± 0.2	0.374
Monoclonal immunoglobulin, n (%)	17 (20)	14 (18)	3 (50)	0.093
**Echocardiography**				
LVEF (%)	52 ± 8.8	52 ± 9.1	52 ± 5.1	0.992
GLS (%)	−12.9 ± 3.6	−13 ± 3.7	−12.1 ± 2.4	0.624
MCF (%)	28.3 ± 13.3	27.4 ± 12	24.1 ± 12.7	0.524
Pericardial effusion, n (%)	10 (12)	6 (8)	4 (67)	<0.001
LVDd (cm)	4.8 ± 0.8	4.8 ± 0.9	4.7 ± 0.5	0.772
RVDd (mm)	4.10 ± 0.73	3.91 ± 0.71	4.5 ± 0.74	0.029
IVSd (mm)	16 ± 2.5	16 ± 2.4	18 ± 3.5	0.075
LVMI (g/m^2^)	150.6 ± 40.3	149.7 ± 40.5	162.1 ± 40.1	0.473
LAVI (ml/m^2^)	55.1 ± 23.4	55.5 ± 24.1	50.2 ± 11.7	0.590
RA dimension (mm)	4.4 ± 0.8	4.3 ± 0.7	5 ± 1.4	0.052
SV (mL)	69.9 ± 23.3	70.1 ± 23.4	67 ± 23.1	0.749
Aortic valve area, cm^2^	0.7 ± 0.2	0.7 ± 0.2	0.6 ± 0.2	0.301
AV MPG, mmHg	39.9 ± 16.7	40.5 ± 17.1	31.1 ± 7.2	0.184
AV Vmax, m/s	4 ± 0.8	4.1 ± 0.8	3.4 ± 1	0.062
E velocity (cm/s)	102.3 ± 40.7	102 ± 41.5	106.1 ± 33.3	0.813
E’ (cm/s)	8.6 ± 2.8	8.6 ± 2.9	8.2 ± 1.5	0.756
Lateral E/e’	11.9 ± 4.7	11.8 ± 4.8	13.6 ± 1.5	0.471
Peak TR velocity, (m/s)	2.9 (2.5–3.4)	2.9 (2.6–3.4)	2.7 (2.3–3.6)	0.557
sPAP (mmHg)	42 ± 15.7	42.1 ± 14.9	41.7 ± 24.2	0.969
TAPSE (mm)	20.2 ± 5.4	20.2 ± 5.5	19.5 ± 4.7	0.749
Right ventricular s’ (cm/s)	10.2 ± 2.1	10.4 ± 2.3	9.4 ± 0.9	0.403
**ECG**				
Atrial fibrillation/flutter, n (%)	29 (34)	28 (35)	1 (17)	0.323
Heart rate (beats/min)	74 ± 12	74 ± 13	72 ± 10	0.693
AV block ≥2 grade, n (%)	17 (20)	15 (19)	2 (33)	0.425
PQ duration, ms *	191 ± 42	188 ± 39	235 ± 69	0.062
QRS duration, ms *	110 ± 36	108 ± 33	139 ± 56	0.038
RBBB, n (%) *	13 (15)	12 (15)	1 (17)	0.780
LBBB, n (%) *	10 (12)	10 (13)	0 (0)	0.252
LAFB, n (%) *	15 (18)	15 (19)	0 (0)	0.154
Low QRS voltage, n (%) *	9 (11)	9 (13)	0 (0)	0.376
Sokolow–Lyon index, mV *	2.1 ± 0.8	2.1 ± 0.8	2.6 ± 0.2	0.259
RWT/SaVR index *	0.07 (0.05–0.1)	0.07 (0.05–0.1)	0.08 (0.06–0.1)	0.398
VMR, mV/g/m^2^ × 10^−2^ *	1.5 ± 0.6	1.5 ± 0.7	1.7 ± 0.3	0.709

AS = aortic stenosis, ATTR-CA = transthyretin cardiac amyloidosis, AV = aortic valve, AV Vmax = aortic peak velocity, BNP = B-type natriuretic peptide, ECG = electrocardiogram, eGFR = estimated glomerular filtration rate, GLS = global longitudinal strain, hs-cTnI = high-sensitivity cardiac troponin I, IVSd = interventricular septum thickness, LAFB = left anterior fascicular block, LAVI = left atrial volume index, LBBB = left bundle branch block, LVDd = left ventricular end-diastolic diameter, LVEF = left ventricular ejection fraction, LVMI = left ventricular mass index, MCF = myocardial contraction fraction, MPG = mean pressure gradient, NT-proBNP = N-terminal pro B-type natriuretic peptide, RA = right atrium/right atrial, RBBB = right bundle branch block, RVDd = right ventricular end-diastolic diameter, RWT = relative wall thickness, SaVR = ECG S-wave from aVR, sPAP = systolic pulmonary artery pressure, SV = stroke volume, TAPSE = tricuspid annular plane systolic excursion, TAVR = transcatheter aortic valve replacement, TR = tricuspid regurgitation, TTE = transthoracic echocardiography, VMR = voltage/mass ratio. * Analysis in patients with non-paced rhythm.

**Table 4 jcm-13-00671-t004:** Scintigraphic imaging findings.

	All n = 85	Lone AS n = 79	AS and ATTR-CA n = 6	*p*-Value
**DPD scintigraphy parameters**				
Perugini grade, n (%)				<0.001
0	76 (89.4)	76 (96.2)	0 (0)	
1	2 (2.4)	2 (2.5)	0 (0)	
2	2 (2.4)	1 (1.3)	1 (16.7)	
3	5 (5.9)	0 (0)	5 (83.3)	
H/CL ratio	1.4 ± 0.6	1.2 ± 0.4	2.7 ± 0.9	<0.001
Radioactivity (MBq)	567.5 ± 83.7	565.8 ± 85.8	589.7 ± 48.9	0.505

AS = aortic stenosis, ATTR-CA = transthyretin cardiac amyloidosis, DPD = ^99m^technetium-3,3-diphosphono-1,2-propanodicarboxylic acid, H/CL ratio = heart-to-contralateral lung ratio.

**Table 5 jcm-13-00671-t005:** Procedural complications.

	All n = 85	Lone AS n = 79	AS and ATTR-CA n = 6	*p*-Value
New-onset conduction disorders warranting pacemaker implantation, n (%) *	8/74 (10.8)	8/70 (11.3)	0/4 (0)	0.332
Access-site bleeding, n (%)	6 (7.1)	6 (7.7)	0 (0)	0.336
Acute kidney injury, n (%)	11 (12.9)	11 (14.1)	0 (0)	0.186
Stroke/TIA, n (%)	1 (1.2)	1 (1.3)	0 (0)	0.699

ATTR-CA = transthyretin-type cardiac amyloidosis, AS = aortic stenosis, TIA = transient ischemic attack, TAVR = transcatheter aortic valve replacement. * No pacemaker prior to TAVR: 74 in total study population, 70 with lone AS and 4 with AS and ATTR-CA.

## Data Availability

Data availability is constrained by the absence of patient approvals, thereby precluding the sharing of research data in compliance with ethical considerations.

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
