# Peer review of "Screening for Occult Transthyretin Amyloidosis in Patients with Severe Aortic Stenosis and Amyloid Red Flags"

_jcm, 2024, doi:10.3390/jcm13030671_

Round 1

Reviewer 1 Report

Comments and Suggestions for Authors

This Study deals with an important and exciting subject 

and very well depicted, 

I congrats the authors for their clear presentation of the results 

The methodology is well constructed but lacks power. 

Major limitations 

Small CA sample

Question 

Can the authors apply the score that already exists? 

Conclusion 

Based on one or two red flags, it is not possible to discriminate CA 

What is the stratification according to the existing scoring system? 

What do the authors think about establishing an additional score with the same parameters? 

The need for a large register to list all possible red flags

Author Response

We would like to thank the reviewers and the editors for their time and the concise review of our work. We have revised the manuscript to address unresolved issues and, below, provide a point-by-point response. All changes in the manuscript are highlighted.

This Study deals with an important and exciting subject and very well depicted. I congrats the authors for their clear presentation of the results. The methodology is well constructed but lacks power.

Major limitations

-Small CA sample

We acknowledge that this constitutes a limitation, and we recognize the importance of conducting larger studies to further validate our findings. We have highlighted this limitation in the Limitations section and discussed potential avenues for future research that include expanding the sample size to enhance the robustness of our conclusions.

Question

-Can the authors apply the score that already exists?

We sincerely appreciate this suggestion. Unfortunately, the requested data is not available to us, as our dataset only includes information from patients with amyloidosis red flags who were specifically included in the study. Validating our findings would require access to data from all patients screened for the study, which, regrettably, is currently unavailable. The validation of existing scoring systems across various patient cohorts of consecutive all-comers is indeed highly essential and would be of great interest in future studies.

Conclusion

-Based on one or two red flags, it is not possible to discriminate CA

We agree with the reviewer. Indeed, based solely on several red flags, discrimination of cardiac amyloidosis can be challenging, as highlighted in our study. Our findings underscore that precise profiling of this dual entity is imperative for identifying specific markers of cardiac amyloidosis. We have emphasized in our manuscript that a comprehensive approach is needed to improve diagnostic accuracy and provide understanding of this dual pathology.

Following changes appear in the manuscript:

Page 12, Lines 364-365:

Unmasking CA in patients with AS using red flags proves to be challenging. A comprehensive approach is needed to enhance diagnostic accuracy.

What is the stratification according to the existing scoring system?

This is a relevant point and we recognize reviewer’s concerns. Unfortunately, due to the very small number of ATTR-AS patients in our study cohort, stratifying our findings is not feasible. However, stratifying our findings and testing our hypotheses in external cohorts of AS patients would be of great interest.

What do the authors think about establishing an additional score with the same parameters?

Establishing an additional score with the same parameters was initially one of the aims of the study, as we hypothesized that the red flag-based amyloidosis screening would detect a much higher rate of dual pathology. Unfortunately, this is not possible due to the very small number of diagnosed cases with cardiac amyloidosis.

The need for a large register to list all possible red flags

Indeed, it is crucial to focus on collecting red flag data in a large, multicenter registry to potentially identify specific markers of cardiac amyloidosis in patients with aortic stenosis.

Reviewer 2 Report

Comments and Suggestions for Authors

This interesting paper describes the role of clinical screening for transthyretin cardiac amyloidosis (CATTR) in patients affected by aortic stenosis. I think that this study was well-written and enough clearly presented. I only have some comments:

Introduction

-it seems quite brief; more detail should be discussed on red flags for amyloidosis. For example, low emphasis was given to polyneuropathy in CA.

Clinical suspect of ATTRv

-there are recent studies employing machine-learning algorhythms to target the clinical suspect before genetic testing for TTR gene. This is interesting because data published point out the role of cardiomyopathy. Reduced emphasis was given on nephropathy and ocular involvement. Probably, it is time to overcome expert opinion and qualitative data to use more standardized protocols to orient the clinical suspicion. Cite and discuss relevant literature.

Clinical assessment

-CTS: was it considered as isolated or bilateral CTS?

-genetic testing was proposed only in the presence of cardiac uptake on scintigraphy. Hence, it is quite limiting to have data from only 5 patients. Indeed, the use of scintigraphy is confirmed for many mutations, but, in some cases, there is a risk of underestimation of hereditary CATTR. As a consequence, it is not correct to exclude CA with scintigraphy for Phe64Leu, or Ser97Phe that often presents false negative. This must be discussed and added to limitations.

Neurological comorbidity

- The authors clearly stated in the text that the actual scenario consists of therapy with neurological indication and prescription. Indeed, therapies with siRNA or ASO for TTR polyneuropathy in mixed phenotypes (see a recent paper on this topic “Use of Drugs for ATTRv Amyloidosis in the Real World: How Therapy Is Changing Survival in a Non-Endemic Area”). Hence, it is crucial a neurological evaluation in all cases of ATTRv at clinical diagnosis and at follow-up to be sure to early treat mixed phenotypes.

Style and grammar

Style is adequate and there are no relevant issues.

Author Response

We would like to thank the reviewers and the editors for their time and the concise review of our work. We have revised the manuscript to address unresolved issues and, below, provide a point-by-point response. All changes in the manuscript are highlighted.

This interesting paper describes the role of clinical screening for transthyretin cardiac amyloidosis (CATTR) in patients affected by aortic stenosis. I think that this study was well-written and enough clearly presented.

I only have some comments:

Introduction

-it seems quite brief; more detail should be discussed on red flags for amyloidosis. For example, low emphasis was given to polyneuropathy in CA.

We appreciate the reviewer’s comment. Screening for amyloid red flags in at-risk cohorts is recommended by international guidelines. However, certain red flags, such as CTS, biceps tendon rupture, or polyneuropathy, may have other etiologies, especially in older patients, reducing their predictive value. We aim to expand the introduction and provide a more detailed exploration of the role of amyloidosis red flags.

Following changes were made in text:

Page 2, Lines 59-65:

“Screening for amyloid red flags in at-risk cohorts is recommended by international guidelines (10, 11). The predictive value of numerous amyloid red flags has been demonstrated in previous studies across various patient cohorts. Conditions such as CTS and biceps tendon rupture are common red flags that have been shown to occur 5-15 years before the manifestation of CA (12, 13). Similarly, peripheral neuropathy is a common neurological disorder with different etiologies and can also present as the initial symptom of amyloidosis (14, 15).”

Clinical suspect of ATTRv

-there are recent studies employing machine-learning algorhythms to target the clinical suspect before genetic testing for TTR gene. This is interesting because data published point out the role of cardiomyopathy. Reduced emphasis was given on nephropathy and ocular involvement. Probably, it is time to overcome expert opinion and qualitative data to use more standardized protocols to orient the clinical suspicion. Cite and discuss relevant literature.

We appreciate this comment. It is unclear whether targeted red flag-based screening of hereditary ATTR would yield a higher ATTR prevalence in this population. Machine learning-based approaches might be helpful in the early detection of CA in patients at risk. We would be pleased to discuss the role of machine learning in the early diagnosis of CA in the Discussion section.

Following changes were made in text:

Page 11, Lines 325-333:

“Recently, machine learning approaches based on clinical, laboratory, and imaging data have increasingly been employed to suspect CA. Machine learning enabled the early identification of ATTRv and helped to identify patients with neuropathy who should undergo genetic testing for ATTRv (35, 36). Furthermore, an ECG-based tool has effec-tively suspected CA with an area under the curve of 0.97 (37). Similarly, a deep learning approach discriminated between amyloidosis and hypertrophic cardiomyopathy (38). Given the increasing number of patients, automated machine learning approaches could potentially be employed for the early identification of patients with cardiac amyloidosis in the future.”

The following references appear in the reference section:

  1. Park JK, Petrazzini BO, Saha A, Vaid A, Vy HMT, Marquez-Luna C, et al. Machine Learning Identifies Plasma Metabolites Associated With Heart Failure in Underrepresented Populations With the TTR V122I Variant. J Am Heart Assoc. 2023;12(8):e027736.
  2. Di Stefano V, Prinzi F, Luigetti M, Russo M, Tozza S, Alonge P, et al. Machine Learning for Early Diagnosis of ATTRv Amyloidosis in Non-Endemic Areas: A Multicenter Study from Italy. Brain Sci. 2023;13(5).
  3. Schrutka L, Anner P, Agibetov A, Seirer B, Dusik F, Rettl R, et al. Machine learning-derived electrocardiographic algorithm for the detection of cardiac amyloidosis. Heart. 2022;108(14):1137-47.
  4. Wu Z-W, Zheng J-L, Kuang L, Yan H. Machine learning algorithms to automate differentiating cardiac amyloidosis from hypertrophic cardiomyopathy. The International Journal of Cardiovascular Imaging. 2023;39(2):339-48.

Clinical assessment

-CTS: was it considered as isolated or bilateral CTS?

Both unilateral and bilateral CTS was considered as red flag.

-genetic testing was proposed only in the presence of cardiac uptake on scintigraphy. Hence, it is quite limiting to have data from only 5 patients. Indeed, the use of scintigraphy is confirmed for many mutations, but, in some cases, there is a risk of underestimation of hereditary CATTR. As a consequence, it is not correct to exclude CA with scintigraphy for Phe64Leu, or Ser97Phe that often presents false negative. This must be discussed and added to limitations.

We agree with the reviewer's observation that false negative scintigraphy, particularly problematic in certain mutations, may contribute to the underestimation of ATTRv, despite its low prevalence in the elderly. We intend to briefly address this limitation in the Limitations section.

Following changes were made in text:

Page 11, Lines 348-351:

“Given the false-negative scintigraphy findings in patients with certain mutations, this could potentially have led to the underestimation of ATTRv, as genetic testing was only performed in patients with positive (Perugini >1) bone scans, despite its low prevalence in the elderly.”

Neurological comorbidity

- The authors clearly stated in the text that the actual scenario consists of therapy with neurological indication and prescription. Indeed, therapies with siRNA or ASO for TTR polyneuropathy in mixed phenotypes (see a recent paper on this topic “Use of Drugs for ATTRv Amyloidosis in the Real World: How Therapy Is Changing Survival in a Non-Endemic Area”). Hence, it is crucial a neurological evaluation in all cases of ATTRv at clinical diagnosis and at follow-up to be sure to early treat mixed phenotypes.

Thank you for your comment. However, discussing the therapies of ATTRv is beyond the scope of our manuscript.

Style and grammar

Style is adequate and there are no relevant issues.

Reviewer 3 Report

Comments and Suggestions for Authors

This is a meticulous study. Its only problem is the very small number of pts with amyloidosis (6!), which however is admitted by the authors. I believe that they should add some comments on the possibility of early treatment of aortic stenosis on preventing amyloidosis.

Also, the possible contribution of new drugs against amyloidosis should be mentioned 

Otherwise, the paper is very well written.

Also I believe that this article should be cited because it offers a new insight: https://www.sciencedirect.com/science/article/abs/pii/S1054880709000350

Author Response

We would like to thank the reviewers and the editors for their time and the concise review of our work. We have revised the manuscript to address unresolved issues and, below, provide a point-by-point response. All changes in the manuscript are highlighted.

This is a meticulous study. Its only problem is the very small number of pts with amyloidosis (6!), which however is admitted by the authors. I believe that they should add some comments on the possibility of early treatment of aortic stenosis on preventing amyloidosis. Also, the possible contribution of new drugs against amyloidosis should be mentioned. Otherwise, the paper is very well written. Also I believe that this article should be cited because it offers a new insight: https://www.sciencedirect.com/science/article/abs/pii/S1054880709000350

We appreciate the reviewers' comments. Reducing amyloid load could potentially play a role, although there is a lack of clear evidence. Certainly, there is a need to identify amyloid even in earlier stages of aortic valve disease, as it might be a risk factor for developing severe AS. Amyloid may contribute to its progression, as demonstrated in this study where valves were found to be involved, suggesting a potential influence on the advancement of AS.

Following changes were made in text:

Page 11, Lines 311-315:

“On the other hand, reducing amyloid load could potentially play a role in the man-agement of AS. Amyloid may contribute to the progression of AS progression, as demonstrated in a study where surgically resected heart valve specimens were analyzed, revealing amyloid deposits in 55% of the specimens, with the highest prevalence ob-served in AS (34).”

Following reference appears in the reference section:

  1. Kristen AV, Schnabel PA, Winter B, Helmke BM, Longerich T, Hardt S, et al. High prevalence of amyloid in 150 surgically removed heart valves--a comparison of histological and clinical data reveals a correlation to atheroinflammatory conditions. Cardiovasc Pathol. 2010;19(4):228-35.

Round 2

Reviewer 1 Report

Comments and Suggestions for Authors

the authors applied a robust methodology with a clearly defined study objective

the conclusions are consistent with the results

the question of detecting ATTR amyloidosis in the presence of aortic stenosis remains open